

# Update on the angular resolution of GRAPES-3 experiment based on Moon shadow analysis

D. Pattanaik[1,2*], S. Ahmad[3], M. Chakraborty[1], S. R. Dugad[1], U. D. Goswami[4],
S. K. Gupta[1], B. Hariharan[1], Y. Hayashi[5], P. Jagadeesan[1], A. Jain[1], P. Jain[6],
S. Kawakami[5], H. Kojima[7], S. Mahapatra[2], P. K. Mohanty[1], R. Moharana[8],
Y. Muraki[9], P. K. Nayak[1], T. Nonaka[10], A. Oshima[7], B. P. Pant[8],
M. Rameez[1], K. Ramesh[1], L. V. Reddy[1], S. Shibata[7], F. Varsi[6] and M. Zuberi[1]

**1** Tata Institute of Fundamental Research, Homi Bhabha Road, Mumbai 400005, India
**2** Utkal University, Bhubaneshwar 751004, India
**3** Aligarh Muslim University, Aligarh 202002, India
**4** Dibrugarh University, Dibrugarh 786004, India
**5** Graduate School of Science, Osaka City University, Osaka 558-8585, Japan
**6** Indian Institute of Technology Kanpur, Kanpur 208016, India
**7** College of Engineering, Chubu University, Kasugai, Aichi 487-8501, Japan
**8** Indian Institute of Technology Jodhpur, Jodhpur 342037, India
**9** Institute for Space-Earth Environmental Research, Nagoya University,
Nagoya 464-8601, Japan
**10** Institute for Cosmic Ray Research, Tokyo University, Kashiwa, Chiba 277-8582, Japan

⋆ diptiranphy@gmail.com

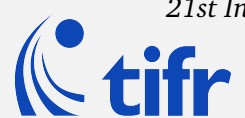

## Abstract

GRAPES-3 is an extensive air shower array located at Ooty (11.6°N, 76.7°E, 2200 m a.s.l.) in southern India. Recently, the angular resolution of the GRAPES-3 array has been improved by correcting the shower front curvature based on the shower size and age. Here, we present the results of the angular resolution and pointing accuracy of the array through observation of the Moon shadow. We have analyzed the data for the period of 2014-2016 containing $\sim 3 \times 10^9$ air shower events with a median energy of 15 TeV. A significant improvement in the angular resolution has been observed compared to the earlier analysis by the group through the Moon shadow method and is comparable to the arrays located at a 2 km higher altitude than the GRAPES-3 experiment.

# 1 Introduction

In the last decade, there has been a significant advancement in the field of very high energy gamma-ray astronomy. Due to the large area and more than 90% duty cycle, the ground based air shower observatories are well equipped to detect the very high energy gamma rays. The flux of the gamma rays is small compared to the large cosmic-ray background. This generally makes the detection of gamma rays challenging. However, improving the angular resolution can reject large cosmic-ray backgrounds. In addition, efficient gamma-hadron separation can lead to a further reduction in the background. With these two methods, ground based experiments are able to detect gamma rays. Large extensive air shower array experiments like Tibet AS$\gamma$ [1], HAWC [2], and LHAASO [3,4] have already implied the presence of Pevatrons (PeV electron accelerators) by detecting gamma rays above 100 TeV up to several PeV.

An excellent angular resolution of the air shower array can be obtained by precise measurement of the arrival time of the air shower. A recent result from the GRAPES-3 experiment has shown that the shower front curvature depends on the shower size and age [5]. If appropriately corrected, the angular resolution can be further improved. The angular resolution can be determined from the Moon shadow method, as suggested by Clark in 1957 [6]. In addition, the Moon shadow method also determines the absolute pointing of the detector, which describes the offset in the location of any point source. In this work, we have observed the Moon shadow in the cosmic rays and determined the angular resolution and pointing accuracy of the GRAPES-3 air shower array.

# 2 The GRAPES-3 experiment

GRAPES-3 is an extensive air shower array located in Ooty (11.4° N, 76.7° E, and 2200 m a.s.l.), southern India. There are 400 scintillator detectors deployed over an area of 25000 m$^2$ [7], and a large area muon telescope (560 m$^2$) [8] is located near the periphery of the array. Fig.1 shows the schematic representation of the GRAPES-3 array with the scintillator detectors arranged in a hexagonal geometry with an inter detector separation of 8 m and the large area muon telescope. The area enclosed within the dashed line is the fiducial area of the array, which covers about an area of 14500 m$^2$. Each scintillator detector is designed to record the charge content as well as the arrival time of the air showers. Recently, the commercially available Time to Digital Converter (TDC) units have been replaced by more advanced High Performance Time to Digital Converter (HPTDC), which exhibits excellent linearity within a dynamic range of 3.5 $\mu s$ with an accuracy of 195 ps. Due to this upgrade, the shower arrival time can be measured more precisely.

# 3 Analysis method

To study the Moon shadow, we used three years of air shower data from 1 January 2014 to 31 December 2016. A total of 2.98×10$^9$ events were recorded with a live time of 994.1 days. The lateral particle density distribution was fitted with the Nishimura-Kamata-Greisen (NKG) function. Only the adequately fitted events were considered for further analysis. In addition to that, another cut was imposed on shower age (defined as the developmental stage of an air shower) to be within 0.2 to 1.8. To remove the contamination due to the large showers falling outside the array, the showers landing within the fiducial area (shown in Fig.1) are selected. The zenith angle of the showers was restricted up to 40°.

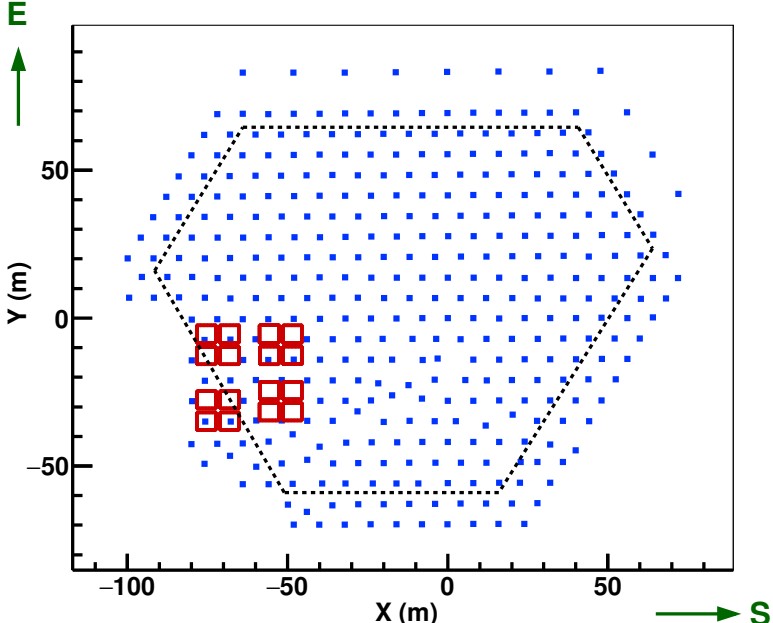

Figure 1: A schematic plot of the GRAPES-3 array showing the scintillator detectors (solid blue markers) and the muon telescope (empty red squares). The dashed line represents the fiducial area of the array covering about 14500 m².

A total of 10 different fake-Moon (background) regions were considered to find the cosmic ray background level. Out of 10 fake-Moon regions, five regions are selected along negative azimuthal ($\phi$) directions, and other regions are selected along positive $\phi$ direction with a 6° successive shift ($\pm 6°, \pm 12°, \pm 18°, \pm 24°, \pm 30°$). The zenith angles ($\theta$) of the fake-Moon regions were kept the same as Moon. The average of the background level was considered to be the reference background. Then the relative deficit in the cosmic rays from the direction of the Moon was calculated using the method explained in [9, 10].

In Fig.2, the cosmic ray flux deficit (%) from the Moon direction is shown as a function of angular distance ($\psi$) from the Moon center. An apparent deficit in the flux can be observed at both $E > 50$ TeV and $E > 100$ TeV. The relative deficit plot was then fitted with a 2-dimensional Gaussian function where the standard deviation ($\sigma$) represents the angular resolution. The 2-d Gaussian expression is given in Eq.(1),

$$N(\psi) = N_0 \frac{\psi_M^2}{2\sigma^2} \, e^{-\frac{\psi^2}{2\sigma^2}} \, , \qquad (1)$$

where $\psi$ is the space angle measured from the center of the Moon, $\psi_M$ is the angular radius of the Moon (0.26°), and $\sigma$ represents the angular resolution of the array. The angular resolution at energy above 50 TeV and 100 TeV was found to be $0.44° \pm 0.07°$ and $0.38° \pm 0.06°$ respectively.

## 4  Results

The angular resolution was obtained at different integral energy bins. The energy dependence of the angular resolution is shown in Fig.3. Obtained values are compared with the earlier Moon shadow analysis done by GRAPES-3 using 2000 to 2003 data [11]. A significant improvement in the angular resolution in the recent analysis can be observed, particularly in low

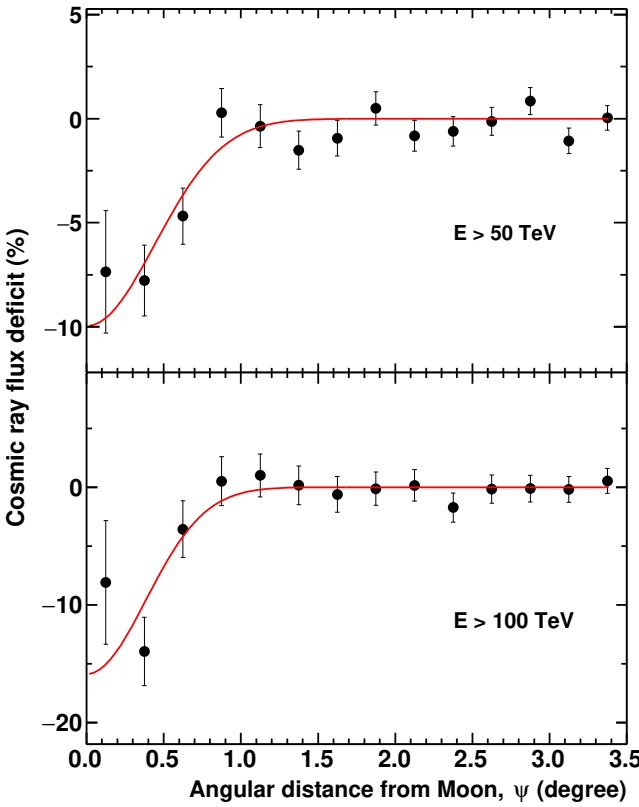

Figure 2: The relative deficit (%) in the cosmic rays flux from the direction of the Moon is shown as a function of angular distance from the Moon center for Energy above 50 TeV and 100 TeV. The deficit plot is fitted with a 2-dimensional Gaussian function to determine the angular resolution.

energies. It has to be noted that the earlier analysis was carried out considering a constant slope value of the shower front curvature, while in this analysis, the shower front curvature was corrected based on the shower size and age. The improved angular resolution agrees with the angular resolution estimated using the Left-Right array division method. In the Left-Right method, the entire array is divided into two sub-arrays based on the line joining the array center and the shower core. By comparing the angle reconstructed from the sub-arrays, the angular resolution is estimated. The detailed method is explained in [5]. This confirms the observation made by the GRAPES-3 experiment that the angular resolution of the array can be improved by the shower size and age based curvature correction.

The other important aspect of studying the Moon shadow is to determine the absolute pointing accuracy of the detector. For this, the GRAPES-3 event coordinates ($\theta$ and $\phi$) were transformed into the equatorial coordinates ($\alpha$ = right ascension and $\delta$ = declination). Using the HEALPix framework, a 2-dimensional relative intensity map was generated between $\Delta\delta$ ($\Delta\delta = \delta_{event}$ - $\delta_{Moon}$) and $\Delta\alpha$ ($\Delta\alpha = \alpha_{event} - \alpha_{Moon}$). By subtracting the Moon coordinates from the events, the position of the Moon was shifted to the center of the map. From the deviation in the observed Moon shadow position from the center, the pointing in $\alpha$ and $\delta$ was estimated [10]. In Fig.4, the relative intensity map is shown, where the blue marker represents the center of the map while the red circle is an ideal representation of the Moon. From the position of the maximum deficit, the pointing in $\alpha$ and $\delta$ was obtained to be $0.032° \pm 0.004°$ and $0.09° \pm 0.003°$.

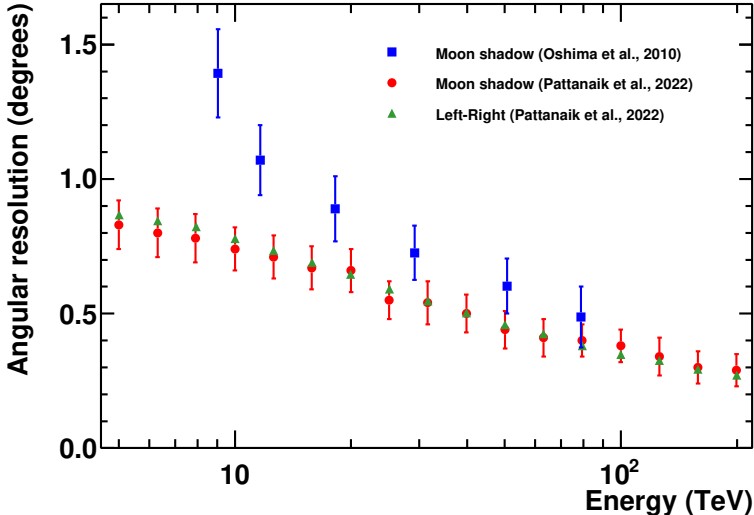

Figure 3: Variation of the angular resolution obtained from the Moon shadow method is shown (red circles) as a function of energy. A clear improvement in the angular resolution can be observed from the earlier Moon shadow analysis (blue squares). The values obtained from the new analysis method are consistent with the angular resolution obtained from the Left-Right method (green triangles).

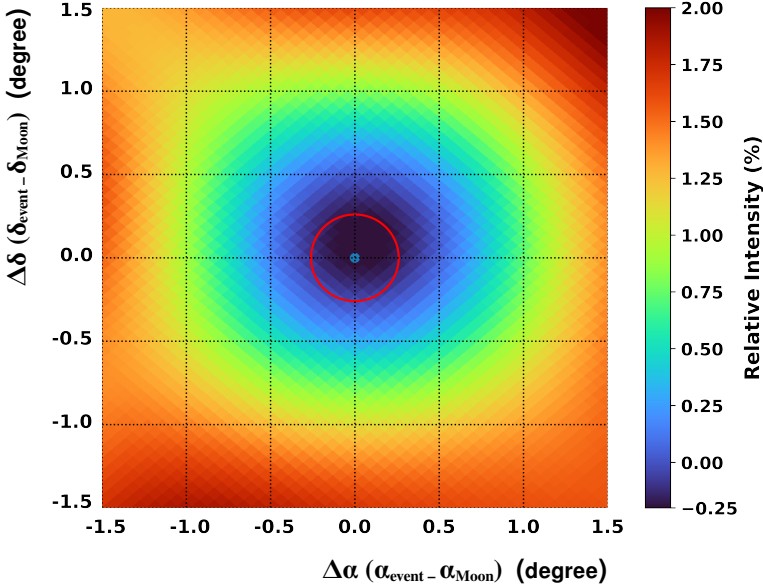

Figure 4: Relative intensity map of the Moon shadow between $\Delta\delta$ and $\Delta\alpha$.

## 5 Conclusion

Compared to the earlier reconstruction method, the new method with shower size and age dependent corrections to the shower front has significantly improved the angular resolution. The angular resolution of the array was obtained to be $0.83° \pm 0.09°$ with a pointing accuracy of $0.032° \pm 0.004°$ and $0.09° \pm 0.003°$ along $\alpha$ and $\delta$, respectively. The pointing accuracy is smaller than the uncertainty in the angular resolution. Even though the GRAPES-3 experiment is located at an altitude of 2200 m, the angular resolution is comparable to the other experiments which are located at twice the altitude of GRAPES-3. In addition to that, the angular

resolution of the array improves to about less than half a degree above 50 TeV energies. Hence the GRAPES-3 experiment is suitable for detecting multi-TeV (E > 50 TeV) gamma rays from the southern as well as northern sky due to its equatorial location.

# Acknowledgements

We are grateful to D.B. Arjunan, A.S. Bosco, V. Jeyakumar, S. Kingston, N.K. Lokre, K. Manjunath, S. Murugapandian, S. Pandurangan, B. Rajesh, R. Ravi, V. Santoshkumar, S. Sathyaraj, M.S. Shareef, C. Shobana, R. Sureshkumar for their role in the efficient running of the experiment.

**Funding information**   We acknowledge the support of the Department of Atomic Energy, Government of India, under Project Identification No. RTI4002. This work was partially supported by grants from Chubu University, ISEE of Nagoya University and ICRR of Tokyo University, Japan.

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
