# Peer review of "Update on the angular resolution of GRAPES-3 experiment based on Moon shadow analysis"

_SciPost Physics Proceedings, doi:SciPost Phys. Proc. 13, 033 (2023)_

## Round 1 · Referee Report · Anonymous (Referee 1) · 2022-10-27

Strengths

The paper shows that the angular resolution can be improved by incorporating more information in the shower reconstruction algorithm.

Weaknesses

The Grammar has quite a few errors, and some explanations are lacking (e.g. left-right method in fig. 3).

Report

In general, it is nice to see the impact of an improvement to the directional reconstruction directly in the moon shadow. This is a fitting topic for a proceeding.

Requested changes

General:

What is the difference between angular resolution and pointing accuracy? This should explained shortly in the introduction, or at the latest in the analysis method section. Currently, there is only a small description in the results section of pointing.

Abstract:

I would leave out the first two sentences.
The angular resolution of the GRAPES-3 array was ... -> Recently, the angular resolution of the GRAPES-3 array has been ..

Introduction:

is tiny -> is small
large cosmic ray -> cosmic-ray
This makes the detection ... -> This generally makes the detection
However, with an excellent ... Sentence sounds wrong here, it should be changed to say something like : However, improvements of the angular resolution can help to distinguish between cosmic rays and gamma rays.

of Pevatron -> of a Pevatron
by detecting the gamma rays -> by detecting gamma rays

The recent result from the GRAPES-3 -> A recent result ...
However, the absolute angular resolution can be determined -> The absolute angular resolution ... (Why "However" here?)

The GRAPES-3 experiment

The GRAPES-3 -> GRAPES-3
Recently the commerically -> Recently, the commercially
the shower arrival time could now be ... -> the shower arrival time can be

Analysis method:

.. imposed on shower age ... What is the definition of "shower age" ?
Fake-Moon -> fake-Moon
were kept the same as Moon -> werre kept the same as the moon

eq 1) Why is the angular moon diameter in the formula and not pi ? Is this explained in ref (9) ? I could not access that reference unfortunately.

Results:

What are "integral energies"?

Fig. 3:
What is the difference between left-right vs the other method? I cannot find a description in the text.

Conclusion:

It should be mentioned, that the angular resolution is improved compared to an older reconstruction that does not take into account shower front curvature corrections.

Hence the GRAPES-3 experiment is
suitable for detecting the multi-TeV .. -> Hence the GRAPES-3 experiment is
suitable for detecting multi-TeV ..

---

## Editorial Decision

published